# Management of Hypertension with Female Sexual Dysfunction

**DOI:** 10.3390/medicina58050637

**Published:** 2022-05-05

**Authors:** Qing Zhong, Yuri Anderson

**Affiliations:** Bioscience Department, Rocky Vista University College of Osteopathic Medicine, Ivins, UT 84738, USA; yuri.anderson@rvu.edu

**Keywords:** female sexual dysfunction, hypertension, antihypertensives, beta-blockers, angiotensin receptor blockers, flibanserin, bremelanotide, ospemifene

## Abstract

Female sexual dysfunction (FSD) in hypertension has been less studied than male sexual dysfunction, and antihypertensive agents’ impact on female sexual function is not defined. In this review, randomized double-blind clinical trials and cross-sectional studies related to female sexual function in hypertension were analyzed from 1991 to 2021. FSD appeared to be higher in hypertensive women than in normotensive women. Beta-blockers are the only antihypertensive agents with relatively strong evidence of damaging the female sexual function. Angiotensin receptor blockers (ARB) are relatively beneficial to female sexual function. To treat FSD in the presence of hypertension, controlling blood pressure is key, and the administration of angiotensin receptor blockers is preferred. In addition to controlling blood pressure, for premenopausal women, flibanserin and bremelanotide can be tried, while ospemifene and hormone supplements are preferred for postmenopausal women.

## 1. Introduction

A healthy sexual function is an important determinant of the quality of life of both men and women. Compared to the well-recognized issue of male sexual dysfunction, less research has been performed on female sexual dysfunction (FSD). While phosphodiesterase 5 inhibitors are successfully used in the treatment of male erectile dysfunction, much less is clear about FSD and its treatments, especially in premenopausal women. Sexual dysfunction has been linked to cardiovascular diseases, such as diabetes, hypertension, obesity, and also with drugs prescribed for cardiovascular diseases [1,2,3,4,5]. As such, hypertension and antihypertensive agents are also reported to increase FSD; however, the results from available research are far from being consistent [1,2,3].

To improve sexual health and quality of life in hypertensive women, it is necessary to understand the relationships between FSD, hypertension, and antihypertensive drugs. In this review, the prevalence of sexual dysfunction in hypertensive women, the possible role of anti-hypertensives in causing FSD as a side effect, and options of treatment for both hypertension and FSD are explored.

## 2. Female Sexual Dysfunction (FSD)

There are similar sexual response cycles in men and women, which include: (1) libido; (2) arousal: vascular dilation, which causes penile or clitoral erection and prostatic or vaginal secretions; (3) orgasm: smooth muscle contraction of the vas deferens during ejaculation or rhythmic vaginal contractions during orgasm in females, and contractions of the somatic pelvic muscles that accompany orgasm in both sexes; (4) resolution [6,7].

Females may encounter decreased libido, difficulty in arousal or orgasm, and pain during sex as symptoms of FSD. According to the most recent Diagnostic and Statistical Manual of Mental Disorder (DSM-5) published in May 2013, there are three types of FSD: 1. female sexual interest/arousal disorder, which includes the previous two categories of female hypoactive desire disorder and female arousal disorder; 2. female orgasmic disorder; 3. genito-pelvic pain/penetration disorder, which derives from previous dyspareunia and vaginismus [8].

FSD is usually investigated by surveying the affected population with several questionnaires. A commonly accepted questionnaire is the Female Sexual Function Index (FSFI), which has 19 questions. These 19 questions are about desire, arousal, lubrication, orgasm, satisfaction, and pain. The highest score is 36, and a score lower than 26.55 indicates FSD [9]. In 2010, for a quick screen, the 19-question FSFI was simplified into a 6-question questionnaire (FSFI-6), which diagnosis FSD with a score lower than 19 [10].

The prevalence of FSD is 41% to 43% in women older than 18 years [11,12]. Although all ages are affected, the experiencing symptoms and contributing factors may differ by age group. In a survey of 1550 women whose ages ranged from 57 to 85 years, 43% of the respondents reported a low sexual desire, 39% had difficulty with vaginal lubrication, 34% had an inability to climax, 17% experienced pain, and 23% reported non-pleasurable sex [13]. A trend of decreased female sexual activity with aging was also described. Sexual activity amongst 57–64-year-old women was 61.6%, while it was 39.5% in women aged between 65 and 74 years, and 16.7% in 75–85-year-old women [13].

The pathophysiology of FSD is correlated with multiple factors, including psychological, interpersonal, vascular, neurological, and hormonal aspects. Psychological factors (including a history of sexual abuse, depression, anxiety) and sociocultural issues play important roles in female sex life. Estrogen is needed to maintain libido and the functioning of the female genital tissue. Low estrogen levels result in vaginal dryness, loss of epithelial cell glycogen, shortening of the vagina, and thinning of the labia [14]. Postmenopausal or hypoestrogenic women are more likely to have dryness of the vagina and reduced sensitivity to touch [7]. In this review, hypertension and hypertensive pharmacologic agents are studied in regard to FSD.

## 3. Hypertension and FSD

To better understand the influence of hypertension and antihypertensive agents on female sexual function, clinical trials and studies were searched in PUBMED from 1990 to 2021 using the keywords “female sexual dysfunction”, “hypertension”, “human”, “antihypertensive”, and “clinical trials”. Four cross-sectional studies (in which drug treatments were enlisted) and six prospective clinical trials were found and are summarized in Table 1 and Table 2, respectively.

The prevalence of FSD appeared to be higher in hypertensive women than in normotensive women. As shown in Table 1, Thomas and his colleagues found that 52.5% of 183 hypertensive women older than 50 and sexually active had FSD [15], a higher percentage than in the general population, in which it was diagnosed in 41–43% of women [11,12,13]. Doumas and his colleagues studied 216 hypertensive women in Greece and found that 47.8% of 136 patients treated with antihypertensives and 32.5% of 80 patients not receiving antihypertensive treatment had FSD as compared to 19.4% of normotensive women [16]. In another study, 90% of 71 hypertensive women had FSD compared to 41% of 85 healthy women [17]. Surprisingly, FSD was observed only in 13.6% of untreated hypertensive women whose ages were between 46 and 50 years, but this incidence rate was still higher than the rate of 4.7% in normotensive women [18]. De Franciscis and his colleagues noticed that FSD incidence was 38% (84/220) in untreated hypertensive postmenopausal women, in contrast to a, incidence of 20% (48/240) in normotensive postmenopausal women [19]. Although one study showed that the prevalence of FSD in hypertensive women was similar to those in the normotensive group [20], a recent systemic review demonstrated that hypertensive women in general have a 1.81-fold relative risk of FSD with respect to normotensive women [21].

**Table 1 medicina-58-00637-t001:** Summary of cross-sectional studies showing the prevalence of FSD and the effect of various antihypertensive agents on sexual function in hypertensive women.

Study	Design	Female Patient Number	Age (y)	Antihypertensive Agents	Women with FSD (%)	Drug Effect on Sexual Function
Thomas, et al., 2016 [15] SPRINT Study	FSFI questionnaire	690 HT (183 sex-active) (19 HT untreated)	>50 Median 67.6	HT treated (BB, ACEI, CCB, diuretics, or multiple) HT untreated	52.5% among 183 sex active patients	None
Spatz, et al., 2013 [20] NSHAP Study	Face to face interview	858 HT treated 234 HT untreated 298 Normotensive	57–85	HT treated (BB, AB, diuretics, ACEI/ARB) HT untreated Normotensive	73.7% 65.3% 71.7%	None
Doumas, et al., 2006 [16]	FSFI Questionnaire	136 HT treated 80 HT untreated 201 Normotensive	Mean 47–48	HT treated (BB, diuretic, CCB, ACEI/ARB) HT untreated Normotensive	47.8% 32.5% 19.4%	Beta blockers predicted FSD
Okeahialam, et al., 2006 [18]	Questionnaire of libido, pain, orgasm	29 thiazide-HT 44 HT untreated 43 Normotensive	Mean 46–50	Thiazide HT untreated Normotensive	17.2% 13.6% 4.7%	Thiazides increased FSD

HT = hypertensive; FSFI = female sexual function index; BB = beta blocker; CCB = calcium channel blocker; ACEI = angiotensin-converting enzyme inhibitor; ARB = angiotensin receptor blocker; FSD = female sexual dysfunction.

Consistent with increased FSD in hypertension, two studies showed that women with controlled hypertension had higher FSFI scores than women with uncontrolled hypertension [16,19]. In Doumas and colleagues’ research, an increase in systolic blood pressure was related to a decreased FSFI score, and the duration of hypertension was negatively correlated with FSFI scores [16]. De Franciscis et al. revealed the highest FSFI score in the patient group with controlled hypertension undergoing therapy, and the lowest FSFI score in hypertensive patients not receiving therapy [19].

FSD in hypertensive women affects all aspects of the sexual response. Nascimento and his colleagues using the FSFI questionnaire found desire disorder (68.2%), excitement disorder (68.2%), lubrication disorder (41.1%), orgasm disorder (55.4%), satisfaction disorder (66.4%), and pain during sex (56.1%) in 157 hypertensive women with an average age of 56 years [22]. In 277 hypertensive Chinese women whose average age was 48, 62.1% of them reported no orgasms [23]. Hypertensive women showed lower vaginal lubrication, less frequent orgasms, and more frequent pain than normotensive women [24].

There are several reasons why hypertension is associated with a higher prevalence of FSD, that may include the following: 1. hypertension causes arterial wall stiffness which reduces vasodilation; 2. hypertension causes endothelial cell dysfunction that reduces nitric oxide (NO) levels, which in turn decrease vasodilation and impact lubrication; 3. hypertension affects the central modulation of sexual behavior; 4. hypertension disrupts the autonomic nervous system [25]; 5. hypertension-related oxidation and inflammation exacerbate female genital tissue fibrosis and damage endothelial cells [26]; 6. hypertension may cause anxiety and depression in some women [22]; 7. antihypertensive drugs may cause sexual dysfunction as a side effect [1,2,3]. It is important to note that the exact mechanism leading to a higher prevalence of FSD in hypertensive women has not been defined.

## 4. Antihypertensive Agents and Female Sexual Dysfunction

Cross-sectional studies on drugs’ effects on female sexual function are summarized in Table 1 [15,16,18,20]. Prospective, randomized, double-blind clinical trials considering antihypertensive drugs and female sexual function are summarized in Table 2. Except for one study that was active-controlled [27], all prospective trials were placebo-controlled [28,29,30,31,32].

**Table 2 medicina-58-00637-t002:** Summary of prospective randomized double-blind clinical trials showing the effect of various antihypertensive agents on sexual function in hypertensive women.

Study	Design	Female Patient Number	Age (y)	Antihypertensive Agents	Female Sexual Function
Ma, et al., 2012 [27]	Active-controlled FSFI	160 HT	18–60	Felodipine–irbesartan Felodipine–metoprolol 48 weeks	Improve Worsen
Van Bortel, et al., 2005 [29]	Crossover Interest in sex	112 HT	Mean 56	Wash out 2 w Placebo 2 w Nebivolol (55) Losartan (57) 12 w	No difference between nebivolol and losartan on libido
Fogari, et al., 2004 [28]	Crossover Questionnaire	120 postmenopausal HT	51–55	Placebo 4w Valsartan Atenolol 16 weeks	Improve Worsen
Grimm, et al., 1997 [31] TOMHS study	Physicians asked about orgasm, frequency of sexual activity	345 HT	45–69	Acebutolol, Amlodipine, Chlorthalidone, Doxazosin, Enalapril Placebo 24 months, 48 months	Amlodipine decreased sexual activity compared to the placebo
Wassertheil-Smoller, et al., 1991 [30] TAIM study	A questionnaire, quality of life, quality of the sex life, frequency of sex, libido, arousal	440 female HT	21–65 Mean 49	Chlorthalidone Atenolol Placebo Low sodium diet Normal diet Weight loss diet 6 months	No difference in FSD between groups
Hodge, et al., 1991 [32]	Crossover Self diary on sexual arousal, desire, and orgasmic function.	18 female HT		Clonidine Prozocin Placebo	Orgasm strength was increased by clonidine compared to the placebo

HT = hypertensive; FSFI = female sexual function index; FSD = female sexual dysfunction.

This section will examine individual antihypertensive agents and their relations to FSD.

**Beta-Blockers**: One cross-sectional [16] and two prospective studies [27,28] showed that beta-blockers significantly negatively affected female sexual function. As shown in Table 1, in Doumas and his colleagues’ cross-sectional study [16], 69.2% of females receiving beta-blockers had FSD, which was significantly higher than 42.7% of patients who did not use beta-blockers. The FSFI scores were negatively correlated with systolic blood pressure, age, and the use of beta-blockers [16]. As shown in Table 2, Fogari and his colleagues performed a crossover trial with 120 postmenopausal hypertensive women. The patients used a placebo for four weeks and then were assigned either atenolol or valsartan in a randomized and double-blinded design for 16 weeks. A survey with 10 questions related to libido, orgasm, and coital activity was carried out. The libido score was decreased by atenolol, whilst it was increased by valsartan compared to basal levels [28]. In another study in 2012 by Ma et al. [27], 160 hypertensive females were treated with either felodipine–irbesartan or felodipine–metoprolol for 48 weeks. The FSFI scores were decreased in the felodipine–metoprolol group compared to the scores before treatment, while felodipine–irbesartan increased the FSFI scores from the basal levels [27], as shown in Table 2. There is evidence that nebivolol, the only beta-blocker, does not reduce hypertensive men’s erection because it increases nitric oxide levels, inducing vasodilation [3]. For comparison, nebivolol’s effect on libido in 55 hypertensive females and losartan’s effect on libido in 57 hypertensive females were investigated in a 12-week treatment. Patients with decreased libido were 30% after using nebivolol vs. a basal level of 34%, while it was 31% after losartan vs. a basal level of 41% [29], as indicated in Table 2. Both nebivolol and losartan did not significantly reduce female sexual desire.

The mechanism by which beta-blockers damage female sexual function is not yet fully understood. Some hypotheses include: (1) beta-blockers can cause sedation and reduce central sympathetic output, which may reduce libido; (2) beta-2 receptor blockage impedes vasodilation; (3) beta-blockers can cause hyperlipidemia and/or hyperglycemia, which exacerbate atherosclerosis. The exact mechanism of beta-blockers’ damage of female sexual function is elusive. Interestingly, patients who were aware of the side effects of beta-blockers tended to report male erectile dysfunction, which however, was reversed by a placebo [33]. A similar phenomenon may be seen in hypertensive females. Women with knowledge of the side effects of beta-blockers may show more symptoms of FSD because of self-suggestion.

However, two prospective studies [30,31] and three cross-sectional studies [15,20,24] did not find any correlation between beta-blockers and FSD. Therefore, prospective clinical trials with a larger number of hypertensive women are needed to confirm the harmful effects of beta-blockers on female sexual function.

**Diuretics**: Okeahialam and his colleagues found that FSD was diagnosed in 17.2% (5/29) of thiazide-treated hypertensive women, 13.6% (6/44) of untreated hypertensive women, and 4.7% (2/43) of normotensive females in a cross-sectional study [18], as shown in Table 1. The main symptom was reduced libido. There was a higher ratio of FSD in the thiazide group, but the difference was not significant, which may be due to the small sample size. However, other studies did not find any relationship between diuretics and FSD, including two prospective randomized double-blind placebo-controlled trials [30,31] and three cross-sectional studies [15,16,20].

Thiazides have been significantly linked to male erectile dysfunction [3,31]. Research has shown that even though the mechanism by which thiazides affect sexual function is unknown, possible causes may be a decrease in blood volume and hyperlipidemia, which aggravate atherosclerosis. It is not certain whether diuretics damage female sexual function currently.

**Calcium Channel Blockers**: As shown in Table 2, in Crimm et al.’s study, 345 hypertensive females were divided into five groups assigned different drugs, each group comprising 50–60 patients; the administered drugs were acebutolol, amlodipine, chlorthalidone, doxazosin, enalapril, in addition to a placebo. The participants were followed for four years [31]. The ratio of decreased sexual activity in the amlodipine group was nearly double compared to that of the placebo group after the two years (15.7% vs. 9%) and four years (26.4% vs. 12.5%) [31]. Although amlodipine had a higher tendency to reduce female sexual activity, this effect was not statistically significant. The lack of statistical significance could be due to the small sample size of only 50–60 subjects in each treatment group [31]. Sexual activity frequency is not included in female sexual dysfunction categories, but a correlation exists between sexual activity frequency and willingness, libido, and satisfaction. This survey only asked two questions, about sexual activity and orgasm, and did not cover the whole sexual scope. In addition, three cross-sectional studies did not find any relationship between calcium channel blockers and sexual function [15,16,20]. Since the cross-sectional studies were retrospective, many patients were treated with multiple antihypertensive drugs for unknown durations, and there was not a placebo-controlled group. Considering this, cross-sectional studies may not convincingly explain the causal effect of the individual drugs. The effect of calcium channel blockers on female sexual function should be investigated further with a prospective trial analyzing a large population of hypertensive women.

**Angiotensin-Converting Enzyme Inhibitors (ACEI)/Angiotensin Receptor Blockers (ARB)**: Two prospective, double-blind, randomized clinical trials, listed in Table 2, support the hypothesis that ARBs improve female sexual function [27,28]. Taking valsartan for 16 weeks increased sex desire and fantasy in 50 hypertensive females [28]. Women treated with losartan for 12 weeks had fewer complaints of “less interest in sex” (30% vs. a baseline of 41%) [29]. In addition, using felodipine–irbesartan for 48 weeks increased the FSFI scores compared to the basal scores [27].

The benefit of ARB drugs on sexual function may be related to vasodilation, an improvement in endothelial function, and a decrease in fibrosis in female genital tissue [1].

However, three cross-sectional studies and one prospective study found no relation between ACEI/ARB and FSD [15,16,20,31]. As stated previously, in cross-sectional studies, most patients used different combinations of antihypertensive agents, so it was difficult to define the side effects of each individual drug. Grimm and his colleagues only investigated orgasm and sexual activity frequency; therefore, they did not consider the whole sexual scope [31].

**Alpha-Blockers**: Prazosin did not change female sexual function [32], but this study was performed only on 18 hypertensive females.

**Clonidine**: It was found that clonidine increased orgasm compared to placebo [32]. Although clonidine is a promising drug to enhance female sexual function, only 18 hypertensive females were examined in this study [32].

From the above-mentioned studies, beta-blockers may be the number one antihypertensive drugs possibly increasing the risk of FSD, followed by thiazide diuretics and calcium channel blockers, while ACEI/ARBs may be beneficial to the female sexual function. Prospective studies are more convincing than cross-sectional studies, but their main limitation is their low sample sizes, as less than 200 hypertensive women were examined in each treated group. With such a small sample size, it is difficult to draw a conclusion about the effects of each antihypertensive on female sexual function, as there may be other confounding issues that could reduce the statistical power of that research. So far, no prospective clinical trial with a large enough sample size has been conducted to investigate the relationship between single antihypertensive drugs and FSD. It is clear that further investigation is warranted examining individual antihypertensive agents and their effects on female sexual function in a large population prospective clinical trial.

## 5. Management of Hypertension in Hypertensive Women with FSD

Women with controlled hypertension had a significantly lower incidence of sexual dysfunction compared to women with uncontrolled hypertension [16,19]; therefore, the best management of hypertension-related FSD should ensure that blood pressure is kept in a normal range. According to the limited information available, ARBs improve female sexual function and should be the preferred pharmaceutical antihypertensive agents. Alpha-blockers and clonidine can be used as well. Nebivolol is another good option. Beta-blockers other than nebivolol may need to be avoided in those women.

## 6. Management of Female Sexual Dysfunction in Hypertensive Women

When female sexual dysfunction develops in hypertensive women, education about the anatomy of the vulvovaginal organ and pelvic floor, psychologic interventions, training of sexual skills, pelvic floor physical therapy, cognitive-behavioral therapy, mindfulness-based therapy, and couple therapy are recommended as non-pharmacological treatments [34]. These therapies should be tried before or combined with pharmacological therapy.

This section described some of the most recent pharmacological therapies for treating FSD. It is important to keep patients informed of the side effects of individual therapeutic agents.

## 7. Pharmacological Treatment of Female Sexual Interest and Arousal Disorder

**(1). Flibanserin**: It is a 5HT_1A_ partial or full agonist, originally an antidepressant, and was approved by the FDA in August 2015 for the treatment of female hypoactive desire disorder (HSDD) in the premenopausal population. Its mechanism involves serotonin, which is an inhibitory neurotransmitter in female sex function, and norepinephrine and dopamine, which are excitatory neurotransmitters. Flibanserin may serve to modulate the balance of these neurotransmitters to improve sexual function. It should be administered at bedtime and stopped after 8 weeks in the absence of any improvement because of side effects, such as somnolence, dizziness, nausea, fatigue, and hypotension [35,36]. It is suggested not to consume alcohol during its administration because of increased hypotension, but a recent study showed that alcohol consumption does not increase the side effects of flibanserin [36].

**(2). Bremelanotide**: It is a melanocortin receptor agonist and was approved by the FDA in June 2019 for the treatment of female hypoactive desire disorder (HSDD) in premenopausal women. It is injected subcutaneously around 45 min before anticipated sexual activity [37,38]. Compared to flibanserin, bremelanotide does not need to be administered daily. Its side effects include nausea, vomiting, flushing, headache, and hyperpigmentation.

**(3). Phosphodiesterase inhibitors and testosterone**: These are used off-label to increase female arousal and libido, respectively. Safety has not been established in older women [14]. Phosphodiesterase inhibitors should not be used together with nitroglycerin, to avoid severe hypotension.

**(4). Under-investigated drug**: Intranasal oxytocin was studied because oxytocin is increased in the brain during arousal [39]. The FDA has not yet approved this drug.

## 8. Pharmacological Treatment of Genito-Pelvic Pain and Penetration Disorders

**(1). Lubricants and moisturizers** including oil, hyaluronic acid-based, and polycarbophil products may help alleviate dyspareunia that is due to vaginal dryness [34].

**(2). Ospemifene:** It was FDA-approved in 2013 for vulvovaginal atrophy due to menopause. It is a selective estrogen receptor modulator (SERM) and exerts a strong, nearly full estrogen agonist effect in the vaginal epithelium [40]. It is administered orally as one tablet daily. It does not increase the risk of thrombosis as other SERM drugs do, such as tamoxifen. Its side effects include hot flashes in 7.5% of patients and mild muscle spasms in 3% of patients [41].

**(3). Hormone supplements**: For menopausal women, oral conjugated estrogen and low-dose vaginal estrogen therapy can be used [34]. In November 2015, the FDA approved intravaginal prasterone (dehydroepiandrosterone (DHEA)) to treat women experiencing moderate dyspareunia due to menopause [42]. Oral conjugated estrogen increases the risk of thrombosis, while thrombosis has not been associated with vaginal estrogen [41]. The most common side effect of intravaginal prasterone is application site discharge [43].

## 9. Conclusions

This review has found that hypertension has a negative influence on female sexual function, and beta-blockers may worsen FSD. It is better to select antihypertensive agents that are relatively safer for female sexual function, such as ACEI/ARB, nebivolol, alpha-blockers, and clonidine, to maintain a good blood pressure control. In premenopausal hypertensive women, flibanserin and bremelanotide can be used to increase libido and arousal. In postmenopausal hypertensive women, hormone supplements and ospemifene might be good options to improve vulvovaginal atrophy. This study found that there is a need for prospective, randomized, placebo-controlled trials in large groups of women to explore antihypertensive agents’ sexual side effects.

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
