# Peer review of "Management of Hypertension with Female Sexual Dysfunction"

_medicina, 2022, doi:10.3390/medicina58050637_

Round 1
Reviewer 1 Report
The authors presented a review regarding the relationship between the Female Sexual Dysfunction (FSD) and hypertension the only antihypertensive agents. They reported that it is better to select antihypertensive agents that are relatively safer for female sexual function.
In addition to controlling blood pressure, for premenopausal women, flibanserin and bremelanotide can be tried, while ospemifene and hormone supplements are preferred for postmenopausal women.
Some notes
Introduction.
- In lines 29, you correctly emphasize that obesity and diabetes are among the various causes. It is therefore appropriate to mention two papers in this regard.
- Diabetes Metab Syndr Obes. 2015 Feb 11; 8:97-101. doi: 10.2147/DMSO.S71376. eCollection 2015. PMID: 25709482
- Obstet Gynecol Clin North Am. 2009 Jun;36(2):347-60, ix. doi: 10.1016/j.ogc.2009.04.004. PMID: 19501318 Review.
-Questionnaire (Lines 53-57): it is worth mentioning that the 6-question questionnaire is also validated.
- J Sex Med. 2010 Mar;7(3):1139-46. doi: 10.1111/j.1743-6109.2009.01635.x. Epub 2009 Dec 1. PMID: 19968774
-Line 89: Doumas et al (reference 13): the percentages reported in the text do not correspond to those reported in Table 1
-Line 109: Change "Franciscis, et al." In "De Franciscis et al .."
-Line 152: Change “Fogaris” to “Fogari…”.
-References in the text: it is better to standardize the citations in the text and then use:
Author et al (more common)
Or: Author and his colleagues
Reviewer 2 Report
I am very interested in the study entitled “Management of Hypertension with Female Sexual Dysfunction” by Qing Zhong. I think there is nothing to point out. .Author Response
Please see attachment

Reviewer 3 Report
Management of Hypertension with Female Sexual Dysfunction
A review of the literature (which is scant) of female sexual dysfunction in regards to hypertension and hypertensive therapies. Other pharmacologic treatment for various forms of female sexual dysfunction were also review. Uncontrolled and untreated hypertension had adverse effects on female sexual function. ARB’s nebivolol, alpha blockers, and clonidine were favorable. The studies were all very small and the paper did a good job at pointing out limitations. Table I and II were difficult to read and interpret and reformatting the tables should be considered.
